# Low-Cost Fiber-Optic Sensing System with Smartphone Interrogation for Pulse Wave Monitoring

**Aleksandr Markvart** [1], **Alexander Petrov** [1]**, Sergei Tataurtshikov** [1]**, Leonid Liokumovich** [1] **and Nikolai Ushakov** [1,2,*]

[1] Institute of Electronics and Telecommunications, Peter the Great St. Petersburg Polytechnic University, 195251 St. Petersburg, Russia; markvart_aa@spbstu.ru (A.M.); petrov.av1@spbstu.ru (A.P.); tataurtshikov.ss@edu.spbstu.ru (S.T.); leonid@spbstu.ru (L.L.)

[2] S.P. Kapitsa Research Institute of Technology, Ulyanovsk State University, 42 Leo Tolstoy Street, 432970 Ulyanovsk, Russia

**\*** Correspondence: n.ushakoff@spbstu.ru

**Abstract:** Pulse wave measurement is a highly prominent technique used in biomedical diagnostics. The development of novel cost-effective pulse wave sensors will pave the way to more advanced healthcare technologies. This work reports on a pulse wave optical fiber sensor interrogated by a smartphone. The sensor performance was tested in terms of signal to noise ratio, repeatability of demodulated signal and suitability of demodulated signals for the extraction of information about direct and reflected waves. The analysis showed that the observed fluctuations of signal parameters are caused by variability of the state of the cardiovascular system and not by the system noise.

**Keywords:** optical fiber sensor; smartphone-based sensor interrogation; pulse wave; cardiovascular monitoring

## 1. Introduction

The Internet of Things and the informatization of society have led to an increasing demand for information. Smart home concepts, industrial automation, the need for telemedicine services, as well as aggravating environmental problems require the early development and implementation of new monitoring and sensing systems [1]. Portable and low-cost sensor systems are of particular importance due to their potential of becoming widespread [2]. In this regard, systems based on smartphones, the research and practical implementation of which began relatively recently, are of great interest from the point of view of the prospects for their potential of measuring capabilities and the practical application for spectrometry [3], chemical sensing [4,5], biosensing [6], healthcare [7], human behavior [8] and many others [9,10]. Despite the relative novelty of such systems, it is already obvious that, due to the computing and telecommuting capabilities of modern smartphones, their use in a variety of measuring tasks presents significant advantages over traditional solutions, primarily due to the significant reduction in the cost of measuring devices implementation and the possibilities of their integration into wireless sensor networks [11,12].

On the other hand, a large number of actual measuring tasks require the use of fiber-optic sensors due to their small size, safety, immunity to electromagnetic interference, biocompatibility, the possibility of combining sensors and building distributed systems, and a wide range of measured influences [13]. There are two types of optical fibers: quartz and polymer. Polymer optical fibers (POFs) based on plastics are particularly suitable for the creation of sensor systems that match the the Internet of Things concept because POF has an ease of operation, low cost, high flexibility, softness, can be used in harsh bending situations, allowing the use of simple plug connectors as well as inexpensive light emission diodes (LEDs) [14–16]. In addition, the lack of the possibility of injury when the plastic

fiber is broken makes it more attractive for healthcare applications, to which many studies are devoted [17–19].

One of the promising methods for health control is pulse wave (PW) monitoring. It is widely used for diagnostics of cardiovascular diseases, diabetes mellitus and other health problems as it provides an accurate and reliable estimate of arterial stiffness [20,21]. The most direct and accurate ways to measure the PW signal and the pulse wave velocity (PWV) is the use of pressure catheters, which, however, is invasive and therefore can not be widespread. Measurement of PW and carotid-femoral PWV is utilized in various commercial devices [22] and is widely investigated in scientific literature using novel sensors, including fiber optic ones [23–29]; however, most of these sensors are based on interference effects, thus requiring complex and expensive interrogation hardware.

Thus, the combination of broad measurement capabilities provided by fiber optic sensors and the computational performance of modern mobile processors is catalyzing the emergence of new diagnostic devices for personal medicine, for inexpensive laboratory biochemical diagnostics, and other measurement tasks where the low cost of the interrogator is extremely important. There are a number of recent works several ways of using only a smartphone for interrogation of optical fiber sensors were proposed [30–39], including biomedical fiber sensors [40–44]. In these systems, the smartphone flash LED is used as a light source, while the smartphone camera acts as a photodetector.

Despite the great interest in this area, a large layer of tasks, the solution of which is necessary to unlock the full potential of using mobile interrogation systems, remains unresolved. Even though similar principles were already used to measure relatively weak signals [45,46], the limitations of the measurement resolution of smartphone-interrogated sensors and ways to improve it have yet to be discovered, despite a great effort towards the noise analysis of camera-based sensors [47–49].

For example, since correct measurement of the pulse wave signal requires high resolution of the sensor, it is not evident if a smartphone-based interrogation of a pulse wave sensor can be implemented to meet the desirable resolution. The current work is aimed at filling this gap by developing a low-cost and portable optical fiber sensing system dedicated to high-quality measurement of pulse wave signals and analyzing the limitations of such systems.

## 2. Sensor Principle

A scheme and a photo of the sensing system proposed in this paper are shown in Figure 1. It consists of a light source, a polymer fiber, a beamsplitter, a transducer and a light detector. In order to reduce the cost and bulkiness of the system, it is possible to use a smartphone flash LED as a light source and a smartphone camera as a light detector. This way no additional energy sources are required for the system to operate. The fibers were fixed in a special holder, manufactured from ABS plastic using a 3D printer. The holder was placed on a smartphone using plastic pegs, visible in Figure 1.

The principle of operation of the system is as follows: the light coming from the smartphone LED is coupled to the input fiber end and is further directed through the fiber, beamsplitter into a transducer, which is applied to the human skin above the artery. In turn, the pulse wave affects the characteristics of the light reflected from the transducer, as shown in Figure 1, inset (a), which is then sent back through the beamsplitter to a smartphone camera.

The intensity-based interrogation is used, which is the simplest and still is able to detect weak signals. Its principle is the following: the light coming from the fiber falls on the diaphragm, which deforms as a response to external perturbation, is reflected from it and is coupled back into the fiber. Since the diaphragm deforms, the distance traveled by light from the fiber to the diaphragm changes, because of this, due to the presence of light divergence and the finiteness of the fiber numerical aperture, there is a change in the intensity of light recorded by the photodetector. Up to an extent, such a sensor structure is equivalent to a Fabry–Perot interferometer; however, since the light is incoherent, the

interference between the waves reflected from the fiber end and the diaphragm does not happen and the change of the overall reflected light intensity is a sum of these two waves' intensities.

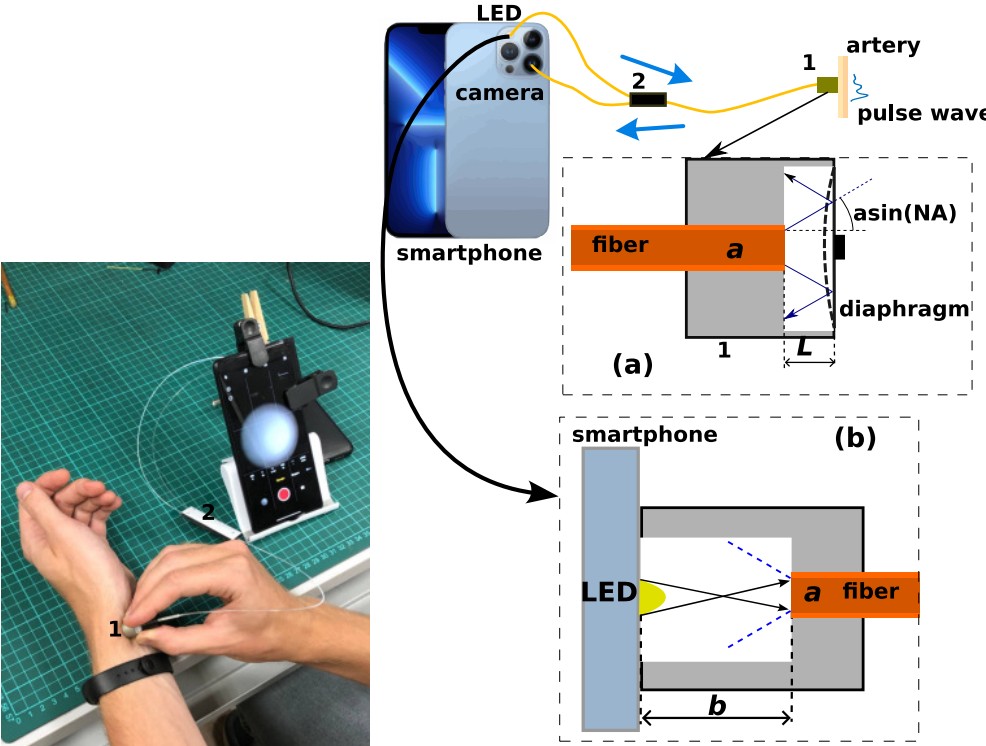

**Figure 1.** Schematic drawing and a photo of a smartphone-based pulse wave sensing system, including a smartphone, a fiber-optic beamsplitter and a transducer. Blue arrows show the direction of light propagation through the fiber scheme. 1—transducer, shown in more detail in the inset (**a**); 2—beamsplitter; inset (**b**) shown light coupling from the smartphone LED to the fiber.

Since spatially incoherent light from the smartphone flash LED is used and the optical fiber has a step-index profile, the divergence of the output beam is mostly governed by the fiber numerical aperture NA, as illustrated in the left-hand side inset in Figure 1 and the portion of the reflected light, coupled back to the fiber is given by the following equation:

$$\eta = \frac{a^2}{\left[a + 4L\tan(\arcsin(\text{NA}))\right]^2},\tag{1}$$

where $a$ is the diameter of the fiber core and $L$ is the distance between the fiber end and the diaphragm.

It should be noted that Equation (1) is valid if all modes of the lead-in fiber are excited, which happens if the input side of the fiber is placed close enough to the light source with a high numerical aperture. Otherwise, if the distance $b$ from the light source to the input fiber end is relatively large, as shown in the inset (b) in Figure 1, only rays traveling with small angles with respect to the fiber axis will be accepted, which reduces the light divergence at the transducer. Black solid lines with arrows in Figure 1, inset (b) show the rays with maximal angles with respect to the fiber axis that are coupled into the fiber. Blue dashed lines in Figure 1, inset (b) show the numerical aperture of the fiber.

Furthermore, the angular spectrum of rays propagating inside the fiber may also be affected by the beamsplitter, used in the current setup to redirect the forward and reflected light to and from the sensing element. Therefore, the overall reflected light intensity will can be expressed as $I_{\text{refl}} = I_0 \cdot (R_0 + R_{\text{mir}}\eta)$, where $I_0$ is the input intensity and $R_0$ and $R_{\text{mir}}$ are reflection coefficients of the fiber end and the diaphragm.

The sensitivity of such a sensor depends on the elasticity of the diaphragm and its dimensions, on the divergence of light in the gap of the sensing element, determined by the numerical aperture of the optical fiber, the shape of the reflecting diaphragm, and on the operating point, that is, on the initial value of the gap between the fiber end and the diaphragm. In order to improve the sensor sensitivity and reduce the dependency of signal quality on transducer placement, a knob, shown in Figure 1, inset (a) was glued to the outer side of the diaphragm. This ensured better mechanical contact between the diaphragm and the tissue.

In this work, a polymer polymethylmethacrylate (PMMA) optical fiber with a core diameter of 900 µm, an outer diameter of 1 mm and a numeric aperture of 0.5 was used. Since the beamsplitters for polymer optical fibers are not widely available commercially, the beamsplitter was manufactured in our laboratory. A 1 × 2 Y-coupler was fabricated with a cutting and gluing technique as shown in Figure 2. Among the many methods of manufacturing such splitters [50–57], this was chosen as one of the simplest and less time-consuming. The first PMMA fiber was cut at an angle of 90 degrees and two others were cut at an angle of 14 degrees. Then, all the fibers were polished with sandpaper and were placed on a 3D-printed housing. After that, they were glued together with a UV-curable glue UV-LOCA TP-2500F, which has high transparency in the optical visible wavelength range, as well as a refractive index of 1.46, close to the refractive index of the fiber core, which is 1.51.

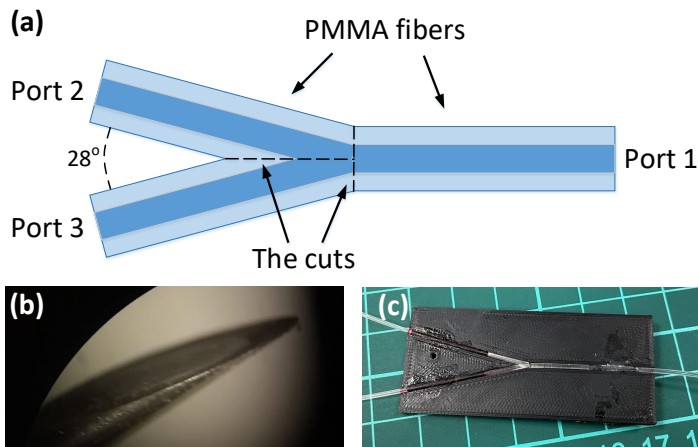

**Figure 2.** Scheme of a 1 × 2 Y-coupler (**a**); photo from a microscope cut at an angle of 14 degrees and polished PMMA fiber (**b**); photo of the assembled Y-coupler before gluing (**c**).

The coupling ratio, insertion and excess loss of this device has been tested using the smartphone flashlight and a photodetector. The coupling ratio of the fabricated sample was 48:52. The insertion loss for transmission from port 1 to port 2 is 6.5 dB, from port 1 to port 3 is 6.4 dB, and from port 2 to port 3 is 32.3 dB. The excess loss is about 3.4 dB.

A Xiaomi Red Note 10 Pro smartphone was used to interrogate the pulse wave sensor. The video was recorded with a 1080p resolution and a frame rate of 60 Hz. Exposure and white balance were fixed, and the focal length was set to the maximum. An example of an image obtained from the camera in case of the proposed sensor connected to the interrogation setup is shown in Figure 3a. A cropped frame is also shown in Figure 3b. In order to present the spatial distribution of light intensity more clearly, pixel values are encoded with a color scheme, shown at the right-hand side of the image.

A special application was developed that controlled the flashlight and the camera settings, performed basic signal processing such as averaging of the selected frame and low pass filtering and was able to save both processed signals and the initial video file for more detailed analysis. The application was developed using Android Studio SDK. Since

the main aim of this paper is an analysis of capabilities of smartphone-based interrogation setup, most parts of the signal analysis were performed on a personal computer.

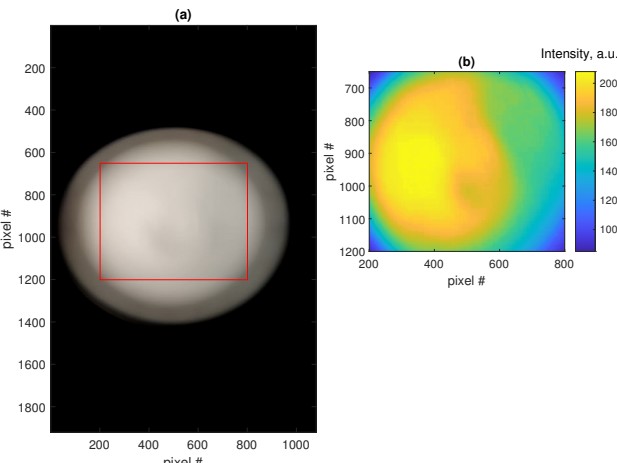

**Figure 3.** Examples of an image, formed by a light, reflected from the proposed fiber optic sensor, full frame, original color (**a**), cropped frame, shown with red rectangle in the full frame, adjusted colormap (**b**).

## 3. Quantization Noises in Smartphone-Interrogated Sensing System

The two main noise mechanisms of the smartphone-based sensing system with intensity readout are light source intensity noises and detector noises, including camera quantization, electronic and thermal noises. Modal noise, present in optical fiber sensors [38] with spectral-domain smartphone interrogation, will have a minor contribution in the considered system due to a great number of modes propagating in polymer optical fiber (hence, small speckle size) and moderate spatial filtering, not causing considerable fluctuations of the received optical power due to the intermode interference.

Light source intensity noise as well as thermal and electronic noises depend on the corresponding hardware and can be estimated experimentally. When a smartphone camera is used as a light detector, thermal and electronic noises of the camera are uncorrelated in different pixels, which means that averaging over the whole image, containing about 1 million pixels, will lead to their great reduction. However, as will be shown in the following analysis, quantization noise demonstrates a more complex behavior. Moreover, the intensity noise of the light source will be correlated in different pixels and thus will not be reduced during averaging.

Obviously, if the target signal is exactly the same in all pixels of the image sequence, then the quantization noises will be absolutely equal as well and hence, will not be suppressed during the averaging. Therefore, there must be some diversity in the averaged signals properties, such as amplitude or mean value so that the quantization noises can be reduced. In the following, we will refer to an intensity variation in a given pixel over time as a partial signal and to a result of averaging of some number of partial signals as a resultant signal.

In order to quantitatively investigate the quantization noise effect, the following numeric modeling was performed. The target signal was assumed to be a harmonic signal with an amplitude varied from 5 to 20 with step 5 at different trials (4 different values), normalized frequency varied from 0.05 to 0.2 with step 0.01 at different trials (16 different values), mean value varied from 0 to 1 with step 0.01 at different trials (100 different values), number of points in the signal equal to 1000. As a result, a $4 \times 16 \times 100 \times 1000$ points array was calculated. Signals with different amplitudes and frequencies were added to the set for better statistical validity. After that, a quantization procedure was applied to this array, performed as rounding towards the nearest integer.

In order to study the effect of averaging on the resulting noise, quantized signals were averaged, where the number of averaged partial signals and the way the signals

were selected from the set were varied. Regular and random selections of signals from the array were used: for regular sampling, some pre-defined *N* number of partial signals with different mean values were extracted from the array and then averaged, the mean values of averaged signals increased uniformly from 0 to 1; for random sampling, the averaged partial signals were extracted randomly, as a result, their mean values were random, uniformly distributed from 0 to 1. Noise level was estimated as standard deviation of difference between the resultant averaged quantized and original signals. Then the mean and the standard deviation of the noise levels were calculated with respect to different signal amplitudes and frequencies. The dependencies of noise levels for regular and random variations of the mean level of signals on the number of averaged signals *N* are shown in Figure 4.

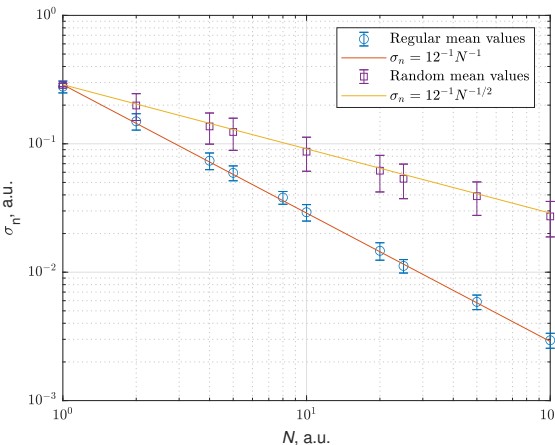

**Figure 4.** Dependencies of averaged signals' noise levels on the number of averaged partial signals in case of regular and random variation of mean value among the signals. Points and errorbars indicate the average and standard deviation obtained in a numeric experiment, solid curves indicate analytic dependencies.

As can be concluded from the dependencies shown in Figure 4, random variation of signals' mean values leads to an anticipated square root dependency of the resultant noise on the number of averaged signals. Since for a single signal the quantization noise standard deviation is equal to $1/\sqrt{12}$, and in case of independent mean values, quantization noises are also independent (which was verified in the performed experiment by calculating correlation coefficients between quantization noises corresponding to different average values, the calculated correlation coefficients did not exceed 0.05), the results of the modeling are in very good agreement with $\sigma_n = 1/\sqrt{12N}$ dependency, shown in the figure.

On the other hand, in the case of regular variation of signals' mean values, reduction of quantization errors becomes much more efficient during averaging, with the resulting noise inversely proportional to the number of averaged signals $\sigma_n = 1/(\sqrt{12}N)$, the corresponding line is also shown in Figure 4.

In practice, partial signals in different pixels can have sufficiently different mean values, having quite complex distributions over the whole image. As a result, practical dependence of the resultant noise on a number of pixels, over which the averaging is performed, can be different from those presented above. Moreover, the contribution of the light source intensity noise can be significant, also affecting the practical correlation properties of partial signals. In the following section, examples of experimentally measured signals and their properties will be presented.

## 4. Measurement Results

### 4.1. Experimental Signals: Noise Analysis

For the measurement, the flash light was switched on, and the camera was operated in fully manual mode in order to adjust exposition, focus, white balance and other parameters and fix them during the signal acquisition.

In order to analyze the noise properties of the obtained signals, pulse wave signals from the carotid artery of Subject 1, measured as described in Section 4.2 were used. The samples of the initial pulse wave signal were calculated by averaging the part of the whole image frame, each frame corresponding to an individual signal sample. Selection of the optimal size of the averaged part of the frame is a nontrivial task and must be specifically investigated. At first, we evaluated the quality of pulse wave signals produced without any averaging. The histogram of SNR values of signals produced from individual pixels (partial signals) is shown in Figure 5a. Interestingly, the spatial distribution of SNR values differs from the intensity distribution, shown in Figure 3b, which indicates that there might be some optimal pattern of pixel averaging, resulting in a maximized SNR value of the averaged signal. SNR was estimated as a ratio of standard deviation of the signal and noise level, which was estimated using median level of signal's FFT according to [58].

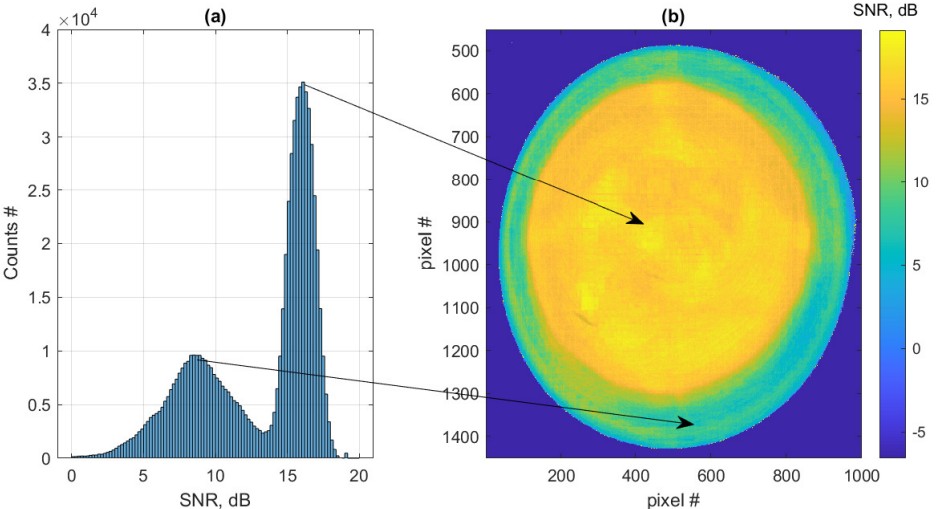

**Figure 5.** Histogram of partial signals' SNR values (**a**) and spatial distribution along the cropped frame of SNR values (**b**). The arrows between the histogram and spatial distribution indicate the correspondence between the spatial areas and the intensity levels.

After that, a set of pulse wave signals was produced by averaging pixels at portions of the whole frame of different sizes. The initial cropped frame had coordinates from pixel #451 to pixel #1450 along the vertical axis and from pixel #1 to pixel #1000 along the horizontal axis, the same as shown in Figure 3. All borders of the other cropped frames were moved inwards with a step of 10 pixels (for the second frame the borders were from pixel #461 to pixel #1440 along the vertical axis and from pixel #11 to pixel #990, and so on). Then, the values of all pixels within these cropped frames were averaged, producing pulse wave signals. For each signal, the signal-to-noise ratio (SNR) was estimated as a metric of signal quality in order to find the optimal frame. The dependency of the SNR value of the averaged signal on the number of partial signals is shown in Figure 6 as circles. Power approximation of this dependency is also shown in Figure 6 as a solid curve.

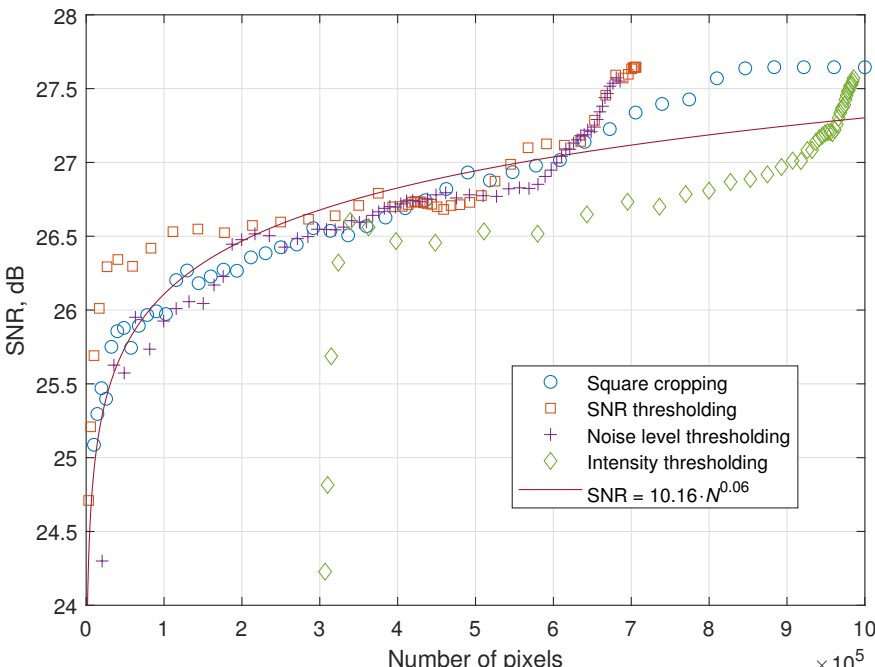

**Figure 6.** Dependency of the averaged signal SNR level on the number of averaged partial signals in case of different approaches of averaging.

As well as that, we have studied how more sophisticated selection of partial single-pixel signals prior to the averaging affects the SNR of the resultant averaged signal. One of the options to perform such partial signal selection is based on the partial signals' SNR value. So, only those partial signals whose SNR values exceeded a certain value were used for averaging, while the signals with lower SNR values were omitted. The dependency of the SNR value of the averaged signal on the partial signals' SNR threshold is shown in Figure 6 as squares. Two other variants of thresholding that were also investigated are thresholding dependent on the noise level of partial signals and thresholding dependent on the mean level of partial signals. In the former method, only those partial signals with noise levels lower than the threshold were used for averaging; in the latter method, all partial signals, which mean values were higher than the threshold level, were used for averaging. In all cases of thresholding, the number of averaged partial signals was calculated for each threshold level and used as arguments for all plots for unification. The shown dependencies can also be interpreted as the SNR value of the averaged signal, obtained in case a given number of partial signals with the highest SNR values (or lowest noise levels, or highest mean values) are averaged.

The power fit of the SNR dependency on the number of averaged pixels is also shown in Figure 6. Its form $SNR = 10.16 \cdot N^{0.06}$ has a weaker dependence on the number of averaged pixels than in the simulation, performed in Section 3. Therefore, it can be concluded that the correlation of partial signal noises is significant, which is most likely due to the influence of intensity noise. The fact that intensity thresholding leads to the most rapid degradation of the resultant signal's SNR value indicates that intensity noises might be suppressed more efficiently by choosing different weighting coefficients for partial signals. Such behavior may also be due to smaller variation of mean values of partial signals in case of their high intensity, which, as shown in Section 3, in turn, leads to less efficient suppression of quantization noises. However, a more detailed analysis of noise in smartphone-interrogated optical fiber systems is out of scope of the current work.

In the current work, the averaging was performed over the whole frame from pixel #451 to pixel #1450 along the vertical axis and from pixel #1 to pixel #1000 along the horizontal axis. However, the dominance of light source intensity noise, revealed in the current analysis, indicates that some modification of the proposed interrogation scheme, allowing one to measure the intensity noise and suppress it, can be used in case higher SNR values are required.

### 4.2. Pulse Wave Signals Processing and Analysis

The developed system was tested through measuring the pulse wave signals at different arteries, namely, radial, carotid and femoral. Four individuals participated in the study: a 29-year-old man (Subject 1), 27-year-old woman (Subject 2), 35-year-old man (Subject 3) and a 29-year-old man (Subject 4). All participants were comprehensively informed about the experimental procedure and gave written consent before the experiment, which was conducted according to the declaration of Helsinki and approved by the institutional ethics committee. For each subject, pulse wave signal measurements were performed in 1 min.

The resulting averaged signal was then filtered using a band pass filter with the cut-off frequencies 0.7 and 15 Hz in order to cancel the low-frequency fluctuations and high-frequency noise. The former may be caused by parasitic environmental changes and prevent correct analysis of the pulse wave signals. The latter degrades the performance of the further processing, including feature extraction. Since the pulse wave signals have very characteristic shapes, unambiguously corresponding to physiological processes, signal features such as systolic peaks, and diastolic dips (often referred to as wave feet) are usually identified in order to simplify the further processing and analysis. In order to accomplish that, a feature detection algorithm [23,59] was applied to the filtered signal. Examples of the pulse wave signals, measured at carotid, femoral and radial arteries of Subject 1 (a)–(d), at carotid artery of Subject 2 (e), at carotid artery of Subject 3 (f) with the extracted systolic peaks and wave feet are shown in Figure 7. As can be seen from the plots in Figure 7, the amplitudes of the pulse wave signals slightly vary, which is caused by different positions of sensors with respect to the arteries for each subject, since the sensing element was placed manually; as well as by different anatomic conditions.

Moreover, an additional test was performed for Subject 4 in order to introduce some diversity of experimental conditions: after being seated with pulse wave signal measured, Subject performed a sit-up exercise for nearly a minute, after which their pulse wave signal was measured again. A comparison of pulse wave signals of Subject 4 before and after the exercise is presented in Figure 8.

The properties of the obtained pulse wave signals were evaluated using the set of metrics proposed in [24]. One of the techniques used in such analysis is fitting the pulse wave signal intervals corresponding to single heart beats by a sum of 6 Gaussian peaks. As a result, 18 parameters, characterizing the direct wave and waves reflected at arterial junctions, are obtained. These 18 parameters turn out to provide information about the state of the cardiovascular system. RPWDF metrics indicate the repeatability of various parameters of the above-mentioned multi-Gaussian fitting model:

- Standard deviations of delays of the first two reflected waves with respect to the direct wave (denoted as $RPWDF_{t_i}$, $i = 1, 2$);
- Standard deviations of widths of the direct and the first two reflected waves (denoted as $RPWDF_{w_i}$, $i$ from 0 to 2);
- Ratio of standard deviations of the first two reflected waves' amplitudes and the corresponding direct wave's amplitude (denoted as $RPWDF_{A_i}$, $i = 1, 2$).

A comparison of the worst and the best metric values is presented in Table 1. It must be noted that a higher signal quality corresponds to higher values of SNR, while CWPI and RPWDF metrics reflect both stability of transducer, variability of cardiovascular system, signal noises and are therefore more complex to interpret.

The diagnostic information about the state of the cardiovascular system that can be extracted from the shape of a single pulse wave signal is the pulse wave velocity, which

depends on the elasticity of arterial walls. Therefore, since the complex shape of the pulse wave signal is caused by reflections of the direct wave, the time delay between these reflected waves has a direct relation with the pulse wave velocity. In turn, the parameters of the multi-Gaussian model can provide information about the pulse wave delay (and hence, pulse wave velocity).

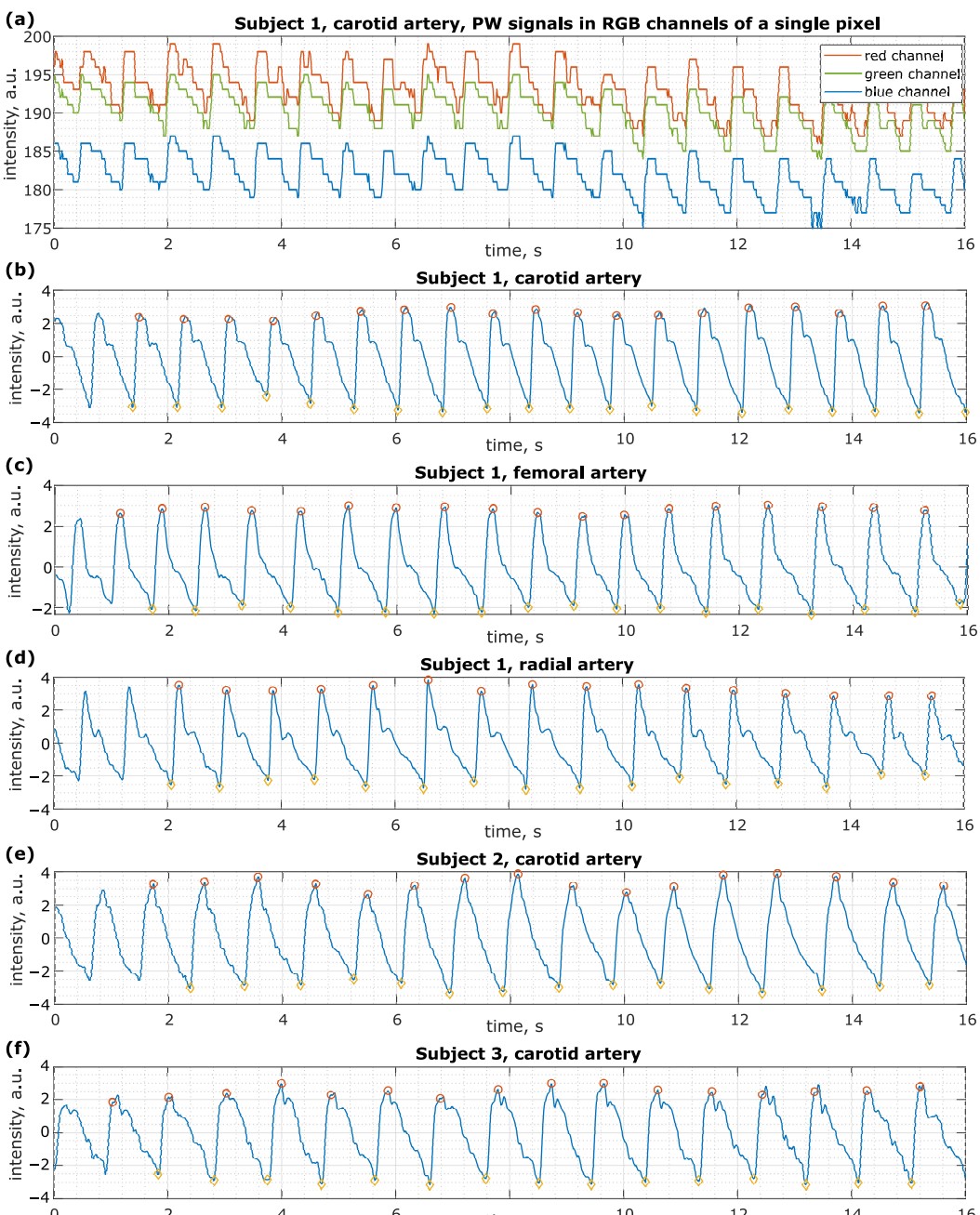

**Figure 7.** Example of the pulse wave signal, which was measured at carotid artery of Subject 1 and registered in RGB-channels of a single pixel (**a**); examples of the averaged and filtered pulse wave signals measured at carotid artery of Subject 1 (**b**), femoral artery of Subject 1 (**c**), radial artery of Subject 1 (**d**), carotid artery of Subject 2 (**e**), carotid artery of Subject 3 (**f**), the estimated positions of peaks and dips are shown with red circles and yellow diamond markers.

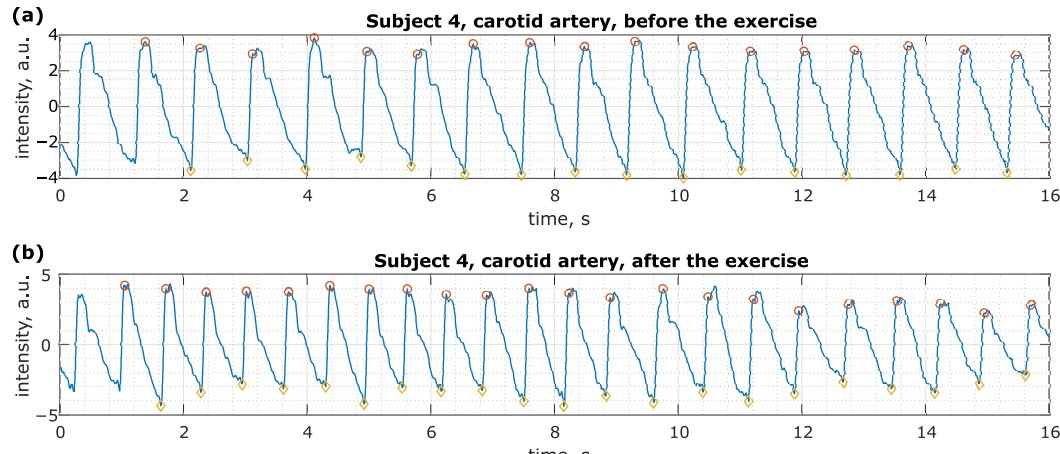

**Figure 8.** Examples of the pulse wave signals, measured at the carotid artery of Subject 4 before (**a**) and after (**b**) a 1 m sit-up exercise. The estimated positions of peaks and dips are shown with red circles and yellow diamond markers.

**Table 1.** Comparison of the best and the worst pulse wave signal characteristics in case of its measurement with the proposed smartphone-based sensor and with a fiber extrinsic Fabry–Perot interferometric pulse wave sensor, as reported in [24]. Please note that for the first two metrics the higher value means better signal quality, while for the rest of the metrics, the lower value means higher signal quality.

| | FPI, [24] | SMP, Min | SMP, Max |
|---|---|---|---|
| SNR, dB | 65 | 20.4 | 25.4 |
| CPWI average, r.u. | 0.91 | 0.93 | 0.99 |
| CPWI $\sigma$, r.u. | 0.07 | 0.007 | 0.1 |
| $RPWDF_{t_1}$, ms | 16 | 6.3 | 17.7 |
| $RPWDF_{t_2}$, ms | 17 | 27.5 | 56.3 |
| $RPWDF_{w_0}$, ms | 4.6 | 3.4 | 6.4 |
| $RPWDF_{w_1}$, ms | 7.5 | 4.7 | 14.3 |
| $RPWDF_{w_2}$, ms | 25.6 | 15.9 | 31.8 |
| $RPWDF_{A_1}$, r.u. | 0.27 | 0.12 | 0.39 |
| $RPWDF_{A_2}$, r.u. | 0.23 | 0.15 | 0.36 |

As was shown in [38], the fundamental limit of estimating the position of a Gaussian peak, determined by Cramer–Rao bound, can be written in the following form:

$$\sigma_\tau^2 \approx 4 \frac{\sigma_N^2}{A^2} \frac{w}{\sqrt{\pi} f_{\text{samp}}}, \tag{2}$$

where $A$ is the amplitude of the Gaussian peak, $\sigma_N$ is the standard deviation of additive noise, $w$ is the width of the Gaussian peak, $f_{\text{samp}}$ is the sampling frequency of the digitized signal. It can be further shown that the lower limit of standard deviation of the Gaussian peak's width is equal to:

$$\sigma_w^2 \approx 2/3 \sigma_\tau^2 = \frac{8}{3} \frac{\sigma_N^2}{A^2} \frac{w}{\sqrt{\pi} f_{\text{samp}}}. \tag{3}$$

Direct estimation of pulse wave arrival time as a position of a systolic peak $\tau_S$ or as a position of the steepest part of the pulse wave rising slope lead to uncertainties due to frequency-dependent transfer characteristics of arterial walls [60,61]. It was shown in [62]

that a correction for these effects can be made by using the following simple equation to estimate the pulse arrival time:

$$\tau_A \approx \tau_S - \sqrt{-\ln(1/2)}\,w. \tag{4}$$

Taking into account Equations (2)–(4), the lower bound of pulse arrival time estimation accuracy is given by the following equation:

$$\sigma_\tau^2 \approx \left[4 - \frac{8}{3}\ln\left(\frac{1}{2}\right)\right]\frac{\sigma_N^2}{A^2}\frac{w}{\sqrt{\pi}f_{\text{samp}}}. \tag{5}$$

For the pulse transit time estimate $\delta\tau = \tau_{A2} - \tau_{A1}$, its variance lower bound can be found as a sum of variances of individual arrival times:

$$\sigma_{\delta\tau}^2 \approx \left[4 - \frac{8}{3}\ln\left(\frac{1}{2}\right)\right]\frac{1}{\sqrt{\pi}f_{\text{samp}}}\left[\frac{\sigma_{N1}^2 w_1}{A_1^2} + \frac{\sigma_{N2}^2 w_2}{A_2^2}\right]. \tag{6}$$

Since the above-mentioned reflections of the direct pulse wave take place in the aorta, the delay between the direct wave and the reflected waves can be used as a measure of the aorta's elasticity. Delays between the direct wave and the first two reflected waves were estimated as differences between the corrected arrival times, which were calculated according to Equation (4). Average delays between the direct wave and the first reflected wave was between 50 ms and 75 ms for different subjects, while standard deviations of these delays were between 10 ms and 21 ms. The delays between the direct and the second reflected waves had an average value between 230 ms and 310 ms and standard deviations between 15 ms and 25 ms. In comparison, Cramer–Rao bounds for pulse delays, estimated according to Equation (6) were in the range between 5 ms and 15 ms, which is slightly lower than the observed standard deviations. Therefore, it can be concluded that the observed variations of the pulse wave delay are caused primarily by physiological causes and not by the system noise.

## 5. Conclusions

In the course of the work, a simple sensing element for measuring the pulse wave signal was created. The sensing principle is based on modulation of the intensity of reflected light. Due to the use of large-core polymer fibers, it was possible to interrogate the sensor with a smartphone, using its LED and camera as a light source and detector, respectively. With the developed system, pulse wave signals were measured on a small sample of subjects. The quality of the measured signals were evaluated using such metrics as signal-to-noise ratio, repeatability of adjacent pulse waves and repeatability of the parameters of pulse wave decomposition into a superposition of Gaussian functions.

Although the signal-to-noise ratio of the pulse wave signals measured with the proposed smartphone-based system is lower than the one of the signals measured with an interferometric sensor, the repeatability metrics are similar and sometimes even superior, indicating the prominence of the proposed pulse wave sensor for biomedical diagnostics. Since smartphone-based interrogation of optical fiber sensors is not only an end in itself, but also can serve as a research platform for developing low-cost interrogation devices, we have also analyzed some general relations related to noise suppression in intensity-interrogated optical fiber sensors.

Moreover, the lower bound of pulse wave delay estimation accuracy was derived according to Cramer–Rao formalism and compared with the experimentally observed variations of pulse wave delay. The performed analysis revealed that the proposed smartphone-based intensity interrogation system for fiber-optic pulse wave sensor is capable of measuring physiological variations of pulse wave signal parameters on a short time interval and hence, can be used for cardiovascular health monitoring.

To the best of our knowledge, it is the first experimental demonstration of pulse wave signal measurement with a low-cost smartphone-based fiber-optic sensing system. In addition, it was revealed that the main contribution to the noise of the measured pulse wave signal is intensity noise of the light source, which can be suppressed by adding a reference optical channel, not modulated by the measured signal, or possibly, by more advanced averaging of the light intensity distribution, captured by the smartphone camera.

**Author Contributions:** Conceptualization, A.M. and N.U.; methodology, A.M. and S.T.; software, A.M. and S.T.; validation, A.M. and A.P.; formal analysis, A.M. and A.P.; investigation, A.M. and S.T.; resources, L.L.; data curation, A.M. and N.U.; writing—original draft preparation, A.M. and N.U.; writing—review and editing, A.P. and N.U.; visualization, A.M. and N.U.; supervision, L.L. and N.U.; project administration, L.L. and N.U.; funding acquisition, N.U. All authors have read and agreed to the published version of the manuscript.

**Funding:** Ministry of Science and Higher Education of the Russian Federation under the strategic academic leadership program "Priority 2030" (Agreement 075-15-2021-1333 dated 30 September 2021), Ministry of Science and Higher Education of the Russian Federation, grant number 075-15-2021-581.

**Institutional Review Board Statement:** The study was conducted according to the guidelines of the Declaration of Helsinki, and approved by the Institutional Review Board of Institute of Electronics and Telecommunications (protocol #2 of 28 April 2023).

**Informed Consent Statement:** Informed consent was obtained from all subjects involved in the study.

**Data Availability Statement:** Obtained experimental data are available from the corresponding author upon reasonable request.

**Conflicts of Interest:** The authors declare no conflict of interest.

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
