# Peer review of "Low-Cost Fiber-Optic Sensing System with Smartphone Interrogation for Pulse Wave Monitoring"

_photonics, doi:10.3390/photonics10101074_

Round 1

Reviewer 1 Report

The work reports on a pulse wave optical fiber sensor, interrogated by a smartphone, although the signal-to-noise ratio of the pulse wave signals measured with the proposed smartphone-based system is lower than the one of the signals measured with an traditional interferometric  sensor, but it is very simple, cost effective, and suitable for common use. What is more, it makes us know smartphone can be used as low-cost interrogation  system for fiber-optic sensor. I think the work is very interesting, innovative and can be accepted. I have some comments as below.

1.   In abstract, the author promoted that the sensor could reduce  susceptibility to motion artifacts, but how does it realize the function for this sensor.

2.     The description of the transduser is not clear, please give more detailed introduction.

3.     In different areas or different time, environment temperature is totally different, how about the environment adaptability of the sensor. This is actually an important problem for the practical application.

Author Response

We would like to thank the Reviewer for useful comments, which helped us improve the manuscript. Below are 

Comment:

1. In abstract, the author promoted that the sensor could reduce  susceptibility to motion artifacts, but how does it realize the function for this sensor.

Reply:

We agree with the Reviewer that the claim of reduced sensitivity of motion artifacts wasn't supported in the manuscript. We have removed this phrase from the abstract. The measure that was taken in order to improve the sensor stability is the knob at the outer side of the sensor diaphragm, which ensured better mechanical contact between the diaphragm and the tissue and decreased the sensitivity of pulse wave signal properties on the placement of transducer with respect to the artery.

Comment:

2. The description of the transduser is not clear, please give more detailed introduction.

Reply:

As was also suggested by other Reviewers, we have revised Figure 1, which depicts the sensing system and the transducer and added some comments about its working principle in Section 2 (lines 78-80, 86-88, 91-93, 97-100).

Comment:

3. In different areas or different time, environment temperature is totally different, how about the environment adaptability of the sensor. This is actually an important problem for the practical application.

Reply:

We agree with the Reviewer. However, due to relatively slow variation of environmental parameters, their influence can be suppressed by a high-pass of band-pass filter of the pulse wave signal, which was done in the presented work.

Reviewer 2 Report

Dear authors,

Development of novel cost-effective pulse wave sensors with increased accuracy and reduced susceptibility to motion artifacts will pave the way to more advanced healthcare technologies. This work is interesting. I have some issues as follows.

a.       Please evaluate the principle of the sensor.

b.       Please evaluate the temperature effect.

c.       What’s the transduser?

Some expression should be rephrased.

Author Response

We would like to thank the Reviewer for useful comments, which helped us improve the manuscript. Below are listed Reviewers comments and our replies.

Comment:

a. Please evaluate the principle of the sensor.

Reply:

We thank the Reviewer for the valuable comment. As was also suggested by other Reviewers, we have revised Figure 1, which depicts the sensing system and the transducer and added some comments about its working principle in Section 2 (lines 78-80, 86-88, 91-93).

Comment:

b. Please evaluate the temperature effect.

Reply:

We thank the Reviewer for the valuable comment. However, due to relatively slow variation of environmental parameters, their influence can be suppressed by a high-pass of band-pass filter of the pulse wave signal, which was done in the presented work.

Comment:

c. What’s the transduser?

Reply:

We thank the Reviewer for the valuable comment. If the Reviewer meant what is the transducer that was used in the presented system, its description is given in Section 2. As described in the manuscript, the operating principle is the modulation of the intensity of light, reflected from the transducer (in other words, sensing element) by the pulse wave. Some clarifications were made in order to better present the material (lines 78-80, 86-88, 91-93).

Comment:

Some expression should be rephrased.

Reply:

We thank the Reviewer for the valuable comment. We have carefully re-read the manuscript and revised some ill-formulated phrases and sentences, which resulted in clearer presentation of the results.

Reviewer 3 Report

The comments about the manuscript are below

  1. Figure 1 needs to be improved for clarification and better reader understanding. The beam splitter and the diaphragm are not indicated in the figure. 
  2. It is not indicated how the fiber was fixed to the camera and flashlight of the used smartphone. 
  3. The insets in Figure 1 need clarification.
  4. What is the difference between the diaphragm and a Fabry Perot interferometer?
  5. Very little is explained about the application that was used on the cell phone to control and process the signal on the smartphone. It needs more explanation. 

writing should be reviewed throughout the manuscript

Author Response

We would like to thank the Reviewer for useful comments, which helped us improve the manuscript. Below are 

Comment:

  1. Figure 1 needs to be improved for clarification and better reader understanding. The beam splitter and the diaphragm are not indicated in the figure.

Reply:

We thank the Reviewer for the valuable comment. As suggested, we have revised Figure 1, which depicts the sensing system and the transducer, indicated the beamsplitter and the diaphragm and added some comments about the working principle in Section 2 (lines 78-80, 86-88, 91-93, 97-100).

Comment:

  1. It is not indicated how the fiber was fixed to the camera and flashlight of the used smartphone. 

Reply:

We thank the Reviewer for the valuable comment. We have added a description of how the fibers were fixed on the smartphone in Section 2, lines 65-66.

Comment:

  1. The insets in Figure 1 need clarification.

Reply:

We thank the Reviewer for the valuable comment. As suggested, we have revised Figure 1, including the insets and added some comments to them in figure caption and in the text (lines 70, 86-88).

Comment:

  1. What is the difference between the diaphragm and a Fabry Perot interferometer?

Reply:

We thank the Reviewer for the valuable comment. In fact, there is no difference, although we avoided using the term interferometer since light with no spatial or temporal coherence was used and no interference could be observed 

Comment:

  1. Very little is explained about the application that was used on the cell phone to control and process the signal on the smartphone. It needs more explanation.

Reply:

We thank the Reviewer for the valuable comment. Since most of the analysis was performed on PC, the main aim of the application was to capture a video file from the camera, control the camera settings and perform the simplest averaging and filtering in order to evaluate the correctness of the transducer placement. We have added some comments on the smartphone application at the end of Section 2 (lines 122-127).

Comment:

writing should be reviewed throughout the manuscript

Reply:

We thank the Reviewer for the valuable comment. We have carefully re-read the manuscript and rephrased some phrases and sentences to improve the English quality. Most of the changes are marked with bold font.